# Nrf2 contributes to the weight gain of mice during space travel

Takafumi Suzuki [1,17], Akira Uruno[1,2,17], Akane Yumoto[3,17], Keiko Taguchi[1,2,4], Mikiko Suzuki [5], Nobuhiko Harada[6], Rie Ryoke[7], Eriko Naganuma[1], Nanae Osanai[1], Aya Goto[8], Hiromi Suda[1], Ryan Browne[7], Akihito Otsuki [2], Fumiki Katsuoka[2,4], Michael Zorzi [9], Takahiro Yamazaki[2], Daisuke Saigusa[2], Seizo Koshiba[2,4], Takashi Nakamura [10], Satoshi Fukumoto[11], Hironobu Ikehata[1], Keizo Nishikawa[12], Norio Suzuki[13], Ikuo Hirano[2,8], Ritsuko Shimizu[2,8], Tetsuya Oishi[13], Hozumi Motohashi [14], Hirona Tsubouchi[15], Risa Okada[3,15], Takashi Kudo[15], Michihiko Shimomura[3], Thomas W. Kensler [16], Hiroyasu Mizuno[3], Masaki Shirakawa[3], Satoru Takahashi [15], Dai Shiba[3✉] & Masayuki Yamamoto [1,2,4✉]

Space flight produces an extreme environment with unique stressors, but little is known about how our body responds to these stresses. While there are many intractable limitations for in-flight space research, some can be overcome by utilizing gene knockout-disease model mice. Here, we report how deletion of Nrf2, a master regulator of stress defense pathways, affects the health of mice transported for a stay in the International Space Station (ISS). After 31 days in the ISS, all flight mice returned safely to Earth. Transcriptome and metabolome analyses revealed that the stresses of space travel evoked ageing-like changes of plasma metabolites and activated the Nrf2 signaling pathway. Especially, Nrf2 was found to be important for maintaining home-ostasis of white adipose tissues. This study opens approaches for future space research utilizing murine gene knockout-disease models, and provides insights into mitigating space-induced stresses that limit the further exploration of space by humans.

---

[1] Department of Medical Biochemistry, Tohoku University Graduate School of Medicine, Sendai, Japan. [2] Department of Integrative Genomics, Tohoku Medical Megabank Organization, Tohoku University, Sendai, Japan. [3] JEM Utilization Center, Human Spaceflight Technology Directorate, JAXA, Tsukuba, Japan. [4] The Advanced Research Center for Innovations in Next-Generation Medicine (INGEM), Tohoku University, Sendai, Japan. [5] Center for Radioisotope Sciences, Tohoku University Graduate School of Medicine, Sendai, Japan. [6] Institute for Animal Experimentation, Tohoku University Graduate School of Medicine, Sendai, Japan. [7] Department of Functional Brain Imaging, Institute of Development, Aging, and Cancer, Tohoku University, Sendai, Japan. [8] Department of Molecular Hematology, Tohoku University Graduate School of Medicine, Sendai, Japan. [9] Department of Molecular Biotechnology and Health Sciences, University of Turin, 10125 Torino, Italy. [10] Division of Molecular Pharmacology & Cell Biophysics, Tohoku University Graduate School of Dentistry, Sendai, Japan. [11] Division of Pediatric Dentistry, Tohoku University Graduate School of Dentistry, Sendai, Japan. [12] Immunology Frontier Research Center, Osaka University, Suita, Japan. [13] Division of Oxygen Biology, Tohoku University Graduate School of Medicine, Sendai, Japan. [14] Department of Gene Expression Regulation, Institute of Development, Aging and Cancer, Tohoku University, Sendai, Japan. [15] Laboratory Animal Resource Center in Transborder Medical Research Center, and Department of Anatomy and Embryology, Faculty of Medicine, University of Tsukuba, Tsukuba, Japan. [16] Translational Research Program, Fred Hutchinson Cancer Research Center, Seattle, WA, USA. [17]These authors contributed equally: Takafumi Suzuki, Akira Uruno, Akane Yumoto. ✉email: shiba.dai@jaxa.jp; masiyamamoto@med.tohoku.ac.jp

During space flight, astronauts experience harsh environments, including microgravity and high-dose cosmic radiation, which affect the homeostasis of physiological systems in our body[1]. To investigate how space flight affects health of animals, space mouse experiments have been conducted exploiting the International Space Station (ISS) and other opportunities[2–10]. However, it has been difficult to attain live return of mice from space or to even realize "space travel" of mice. For instance, the Italian mice drawer system housed six male mice individually, but more than half of the animals died during habitation in space[8–10]. Many of the other preceding space mouse projects did not attempt live return of the mice from space.

To achieve live-return of space mice, the Japan Aerospace Exploration Agency (JAXA) recently established a fully-equipped mouse experimental system for space flight[11]. It comprises mouse habitat cage units (HCU), transportation cage units (TCU), and a centrifuge-equipped biological experiment facility (CBEF). These apparatuses in combination can house mice individually and realize artificial gravity experiments in space. In the first and second missions utilizing this equipment, referred to as Mouse Habitat Unit-1 and -2 (MHU-1 and MHU-2 conducted in 2016 and 2017, respectively), 12 wild-type (WT) male mice were successfully launched each time, and all the mice returned safely to Earth after approximately 1 month stays in the ISS[11,12]. One salient finding in these missions is that ageing phenotypes, such as reduction of bone density and muscle mass, were markedly accelerated during space flight; notably, these phenotypes could be prevented by housing in space with artificial gravity.

It has been shown that various environmental stresses, including oxidative and toxic chemical (often electrophilic) stresses, activate Nrf2 and downstream signaling pathways[13]. Nrf2 is the master transcription factor mitigating oxidative stress. Nrf2 induction is well-known to prevent various diseases, including cancer, diabetes, and inflammation[13]. As cosmic radiation induces oxidative stress[1] and mechanical stresses can also induce Nrf2 activity[14], we hypothesized that space stresses may activate the Nrf2 signaling pathway allowing, in turn, for Nrf2 to play important roles in regulating adaptive responses to these space stresses. Although one preliminary analysis exploiting astronaut's hair samples during space flights showed that expression of NRF2 was decreased in their hair roots during space flight[15], the experimental design harbored limitations. Therefore, we decided to address further studies that would provide direct lines of evidence on this point.

In order to test the hypothesis that Nrf2 contributes to the adaptive maintenance of animal homeostasis in response to the stresses of space flight, it was critical to establish space travel of *Nrf2* gene-knockout model mice[16]. To do this, however, we needed to overcome various challenges, both technical and regulatory, as space travel of gene-modified disease-model mouse lines coupled with a need for live-return had never been conducted. To this end, we decided to utilize the MHU technologies and proposed in 2016 an experiment for JAXA to send Nrf2-knockout (Nrf2-KO) mice to space.

In this regard, metabolite concentrations of body fluids are considered as quantitative traits that can describe a real-time snapshot of physiological state of animals[17,18]. However, little is understood about the response of plasma metabolites to the space stress. Due to the constraints of space experiments, there had been no attempt that focused on changes of metabolites in onboard and post-flight rodent blood samples. Recently, metabolome technologies based on mass-spectrometry have achieved the sensitivity required for analyses of very small amounts of blood[19]. Moreover, metabolite profiling by nuclear magnetic resonance (NMR) has also become a precise and reproducible method for biomarker discovery[17,18].

To elucidate roles that Nrf2 plays in regulating adaptive responses to space stresses, in this study we conducted the one month-long space travel of six Nrf2-KO mice and six WT mice. After a 31-day stay in the ISS in 2018, all of the flight mice returned safely to Earth. Using metabolomic as well as transcriptomic, histological, morphometric, and behavioral methodologies, we found that Nrf2 signaling was indeed activated by space stresses in various tissues. Space stress and Nrf2-deficiency brought about changes in gene expression and plasma metabolite profiles independently for the most part, but cooperatively in certain situations. In particular, Nrf2 is important for the weight-gain of space mice and maintenance of white adipose tissue homeostasis in response to the stresses of month-long space travel.

## Results

**Outline of MHU-3 mouse project utilizing Nrf2-KO mice.** To study contributions of Nrf2 to the protection of mice against the stresses of travel to and from space, and maintenance of homeostasis during their space stay, we have conducted the MHU-3 project. Male Nrf2-KO mice[16] and WT mice were bred and selected for space travel. For this purpose, 60 WT and 60 Nrf2-KO mice at 8-week-old were delivered to the Kennedy Space Center (KSC) 3 weeks prior to launch. These mice were acclimatized to individual housing cages. After acclimation, we selected 12 mice for flight to the ISS based upon their body weight, levels of food consumption and water intake and the phenotypic absence of a hepatic shunt (see "Methods").

SpaceX Falcon 14 rocket (SpX14) containing the mice in the TCU within the Dragon capsule was launched on April 2, 2018 (GMT) from KSC (Fig. 1a). After arrival at the ISS, mice were relocated to the HCU by the crew. We used an HCU that accommodates one mouse per cage[11,12]. Before mice were returned to Earth, they were transferred back into the TCU and loaded into the Dragon capsule which subsequently splashed down in the Pacific Ocean offshore from Southern California on May 5. The TCU was unloaded from the capsule and transported to Long Beach Port on May 7. All mice were alive upon return and were then euthanized and dissected at a laboratory to collect tissues after a general health assessment and a series of behavioral tests.

A ground control (GC) experiment precisely simulated the space experiment was conducted at JAXA Tsukuba in Japan from September 17 to October 20, 2018. Six WT and six Nrf2-KO mice were individually housed in the same units as the flight experiment (FL). Both the TCU and HCU were placed in an air-conditioned room with a 12-h light/dark cycle. Fan-generated airflow (0.2 m/s) inside the HCU maintained the same conditions as in the FL setting.

During the onboard habitation, the health conditions of each mouse was monitored daily by veterinarians on the ground via downlinked videos (Yumoto et al., in preparation). Representative images of all 12 mice in the HCU and their onboard movies are shown in Fig. 1b and Supplementary movie 1, respectively. During the flight mission, temperature, and humidity were well controlled, and concentrations of carbon dioxide and ammonia were maintained at low levels[11]. The absorbed dose rate of radiation was 0.29 mGy/day during the flight.

A new device was developed to collect peripheral blood from the tail in space (Fig. 1c). We obtained approximately 40-μL blood from each mouse with minimal hemolysis (Fig. 1d). Blood collections from tail veins were performed the same way as in the space before launch and after return to Earth, as well as in the GC experiment.

Initial inspection of mice returned to Earth showed loss of balance and enfeebled muscle power (Supplementary Movie 2).

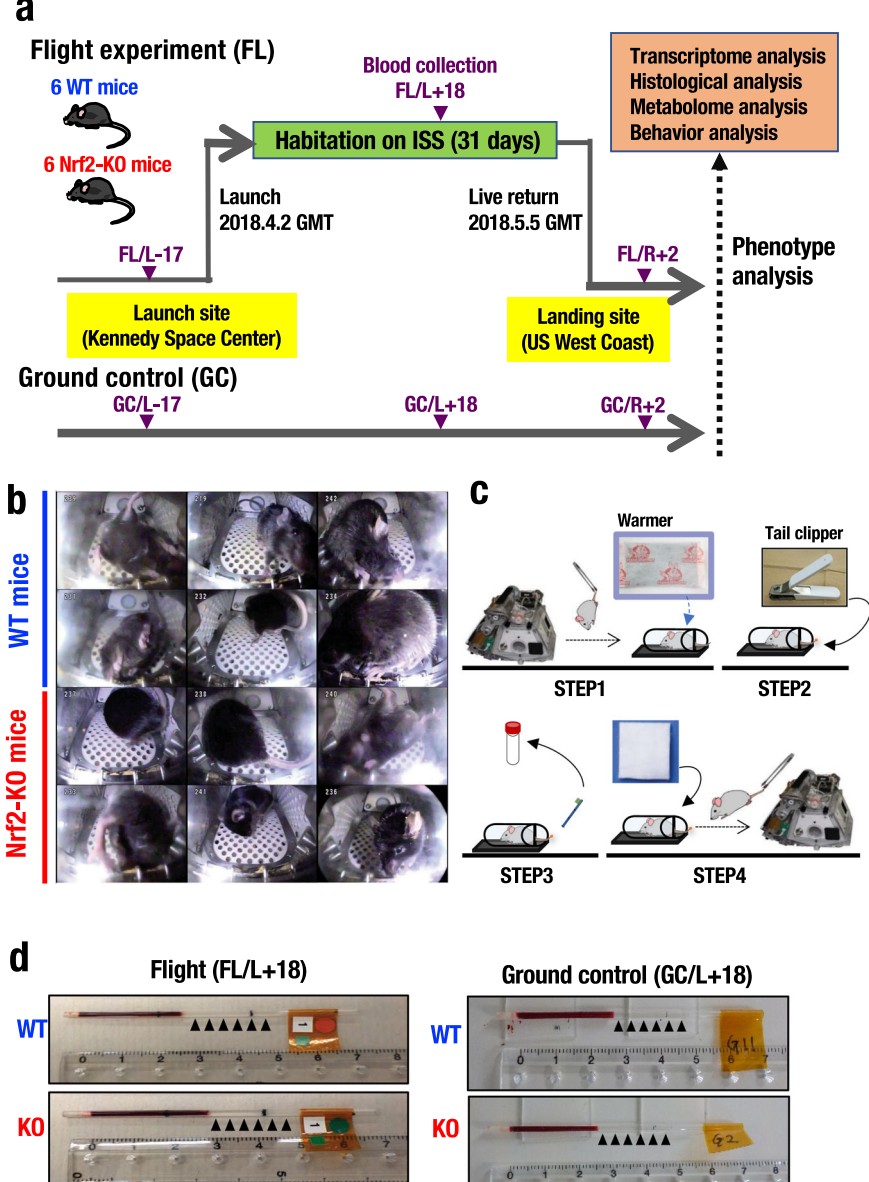

**Fig. 1 The JAXA mouse project using Nrf2-KO mice. a** Overview of the MHU-3 study. Six wild-type mice (WT) and six Nrf2-KO mice (KO) in the C57BL/6J background were launched by SpX14 from the Kennedy Space Center in Florida on April 2, 2018. After 31 days of habitation on the ISS, the mice splashed down in Pacific Ocean near Southern California on May 5. The mice were transported to a laboratory for behavioral observations and the other analyses two days later. **b** Representative images of onboard habitation for WT and Nrf2-KO mice. **c** Overview of the blood collection procedure. All steps are carried out while in orbit: (1) transfer each mouse to a restraining device (covered by a disposable heat pad prior to transfer); (2) use a tail clipper to cut off a 1 mm tip of the mouse tail; (3) collection of blood into a capillary tube inserted into a centrifuge tube, followed by centrifugation; and (4) hemostasis of the tail by applying pressure using a hemostat. **d** Representative pictures of blood samples of flight and ground controls. Arrowheads indicate the plasma fraction with the absence of hemolysis.

The Nrf2-KO flight mouse No. 4 (FL04) developed severe intestinal hemorrhage during the return flight for an unknown reason. Data from FL04 were deemed outliers for many measurements; therefore, data for FL04 were omitted for most of the analyses.

**Space stresses activate the Nrf2 signaling pathway.** To examine whether space stresses activated Nrf2 and downstream pathways, we conducted a wide-ranging RNA-sequence analysis. For this purpose, we dissected the space mice soon after their return to Earth and prepared RNA samples from many tissues, including temporal bone (TpB), mandible bone (MdB), spleen (Spl), liver (Liv), epididymal white adipose tissue (eWAT), inter-scapular brown adipose tissue (iBAT), thymus (Thy), kidney (Kid), and brain cerebrum (Cbr) (Fig. 2a). We selected 26 genes that encode enzymes for detoxication and antioxidative responses that are well-known Nrf2 target genes[20]. The expression of these typical Nrf2 target genes was upregulated widely in various tissues of the FL_WT mice compared with those in GC_WT mouse tissues (Fig. 2b).

In contrast, expression of these 26 selected genes were markedly lower in tissues of the GC and FL Nrf2-KO mice (Fig. 2b), indicating that the upregulation of these genes was attributable to the functional presence of Nrf2 signaling. By comparing FL_KO with GC_KO, we examined all the gene expression changes induced by space flight in the WT mice. The

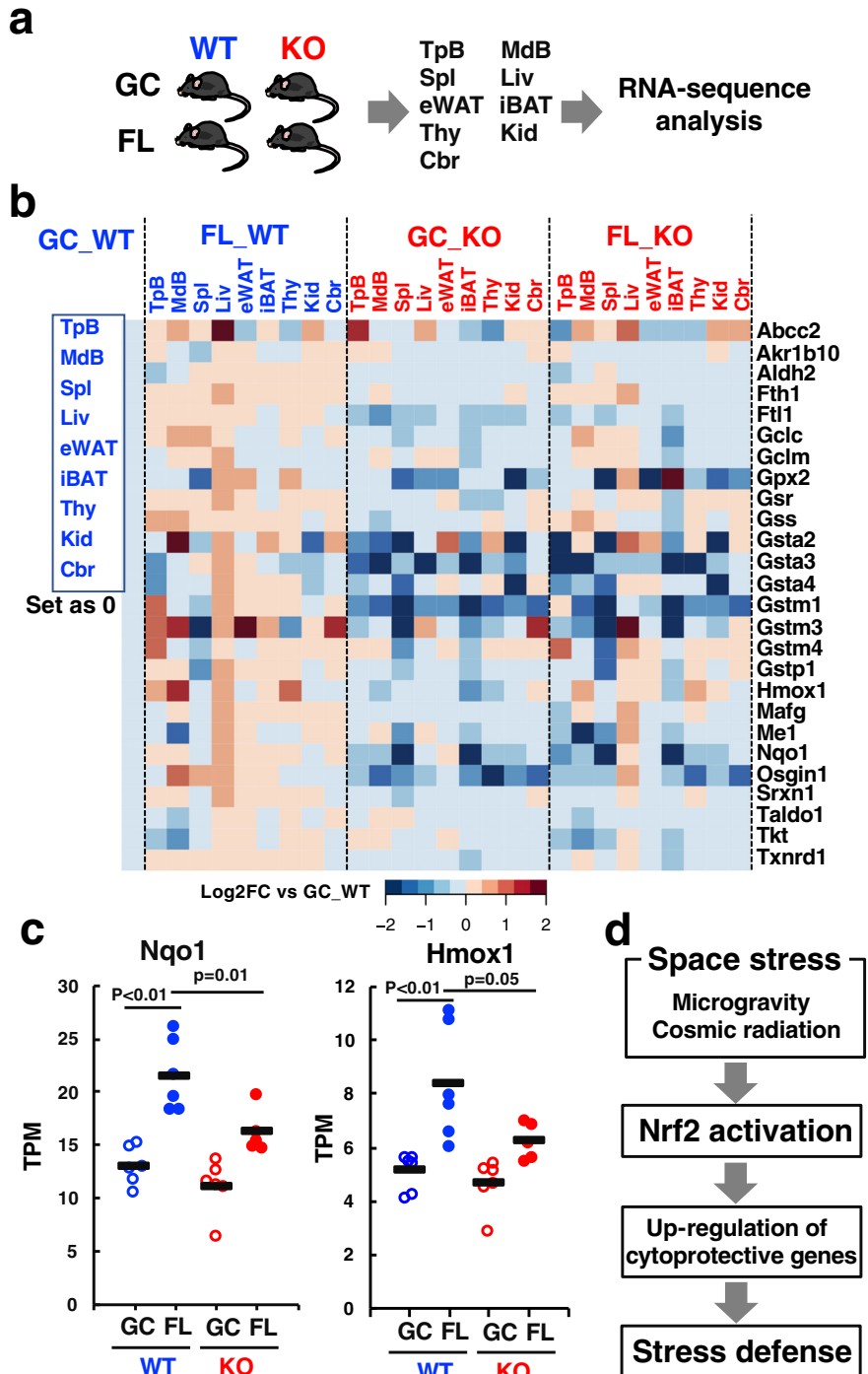

**Fig. 2 Space stresses activate the Nrf2 pathway. a** RNA-sequence analyses of tissues from WT and Nrf2-KO mice in GC and FL groups. **b** Heatmap of expression levels of representative Nrf2-target genes in temporal bone (TpB), mandible bone (MdB), spleen (Spl), liver (Liv), epididymal white adipose tissue (eWAT), inter-scapular brown adipose tissue (iBAT), thymus (Thy), kidney (Kid), and brain cerebrum (Cbr) from WT and Nrf2-KO mice in GC and FL groups. **c** Expression levels of Nqo1 and Hmox1 genes in thymus from WT and Nrf2-KO mice in GC and FL groups. Data are presented as mean, and dots represent individual animals. *n* = 6 for GC_WT, FL_WT and GC_KO, and *n* = 5 for FL_KO, one-way ANOVA with Tukey–Kramer test. **d** Model for space stress response system by Nrf2 signaling.

results of gene set enrichment analysis (GSEA) suggest that a number of pathways are affected by space flight (Supplementary Table 1). The heatmap analyses revealed that space flight-induced changes of gene expression in various tissues could be classified into a Nrf2-dependent group and a Nrf2-independent group (Supplementary Fig. 1a–e). Of note, Nrf2-dependent space-induced genes include typical Nrf2 target genes. Closer inspection

of the extent of gene expression reduction revealed that the response was weaker in FL_KO mice than in GC_KO mice. We surmise that that activation of other stress response pathway(s) by the strong space stresses might alter the magnitude of repression (or loss of constitutive expression) elicited by the *Nrf2* gene knockout in the space flight mouse tissues. However, it should be noted that the Nrf2 pathway contribution is the strongest among

the regulatory pathways for the expression of these genes, as reductions of expression were substantial in both GC and FL Nrf2-KO mice compared with both GC and FL WT mice.

We extended these heatmaps to direct measurements, and observed significant induction of transcriptional expression of Nqo1 and Hmox1 in thymus from FL_WT mice (Fig. 2c). In very good agreement with the heatmaps, in some of the tissues of FL_KO mice, the magnitudes of lower expression were dampened, but still the space-flight induced increase of Nrf2-target gene transcripts were largely canceled in the knockout mice. These results unequivocally demonstrate that Nrf2 activity is indeed induced during space flight and enhances the expression of cytoprotective genes, strongly arguing that our body exploits the Nrf2 signaling pathway to counteract space stresses (Fig. 2d).

**Space flight and Nrf2-deficiency influenced gene expression**. To further clarify contributions of Nrf2 to gene expression patterns of the space mouse, we performed principal component analyses (PCA) of the transcriptomic results. We utilized all identified transcripts for this analysis. Results of the PCA revealed that space travel and Nrf2-deficiency both influenced gene expression profiles, resulting in four distinct patterns. In thymus and eWAT, PC1 separated FL vs. GC, while PC2 separated Nrf2-KO vs. WT (Fig. 3a, b). By contrast, in liver and spleen, PC1 separated Nrf2-KO vs. WT, while PC2 separated FL vs. GC (Fig. 3c, d). These results delineate the first and second patterns in which both space stresses and Nrf2-deficiency strongly and independently elicited specific changes in gene-expression profiles.

A third pattern was found in iBAT in which PC1 separated FL vs. GC, while PC2 did not separate the other groups at all (Fig. 3e), indicating that space travel influenced the gene expression in this tissue while Nrf2-deficiency did not. We found a fourth pattern in Cbr. Somewhat to our surprise, there was no clear separation of gene expression patterns by PCA in Cbr (Fig. 3f), indicating that neither space stresses nor Nrf2-deficiency affected gene expression in this tissue when analyzed en bloc. We surmise that this pattern reflected the cell heterogeneity of Cbr. Collectively, these four PCA patterns of transcriptome analyses demonstrated that, in most of the cases, space stresses influenced gene expression differently than Nrf2-deficiency.

**No acceleration of bone/muscle degeneration in Nrf2-KO mice in space**. Recent MHU1 and MHU2 reports revealed that ageing phenotypes, such as reduction of bone density and muscle mass, were accelerated during the month-long space flights[11,21]. Our results nicely confirmed these observations. Micro-computed-tomography images clearly showed decreased bone density of FL_WT and FL_KO mouse bones compared with GC_WT and GC_KO mouse bones (Supplementary Fig. 2a, b). Against our expectation, Nrf2-gene knockout did not accelerate the decrease of bone mineral density (BMD) during space travel. Muscle mass of soleus (Supplementary Fig. 2c) and gastrocnemius muscles (Supplementary Fig. 2d) also showed substantial decreases during space travel, and again loss-of-Nrf2 did not accelerate nor decelerate these declines. These results indicate that Nrf2-deficiency did not influence substantially the progression of these ageing phenotypes in space. We envisage that the space stress-originated phenotypes of bone and muscle in the mice are very strong, whereas baseline expression levels and impact of Nrf2 are low, so that any possible Nrf2 contribution could not be visible in this context.

**Space flight induces ageing-like changes**. The observation that there was no apparent acceleration of bone and muscle changes during space flight of Nrf2-KO mice compared to FL_WT led us

to examine changes in plasma metabolites associated with ageing. Herein, we addressed whether ageing-related changes of metabolites were accelerated by the deficiency of Nrf2-regulated cytoprotective systems. To this end, we collected blood plasma from the mouse inferior vena cava soon after the return of the mice and conducted NMR-based metabolome analyses (Fig. 4a). Whereas PCA of the metabolome results did not show strong separations compared to that of transcriptome, closer inspection indicated that PC1 separated FL_KO vs. FL_WT (Fig. 4b), suggesting that both space flight and Nrf2-deficiency contributed to changes in plasma metabolites.

We then searched for plasma metabolites, which changed only by space flight or changed by both space flight and Nrf2-deficiency. Of all 40 metabolites examined by NMR-based metabolome analyses (Fig. 4c–e, Supplementary Figs. 3 and 4), we identified three metabolites of interest; i.e., glycerol, glycine, and succinate (Fig. 4c–e). Plasma glycerol levels were increased by flight in both WT and Nrf2-KO mice compared to respective GC mice and with little influence of Nrf2-deficiency (Fig. 4c). Plasma levels of glycine (Fig. 4d) and succinate (Fig. 4e) in FL_WT mice were much lower than GC_WT mice. Glycine and succinate levels in GC_KO mice were also much lower than those in GC_WT mice, and levels did not decrease further with space travel. We additionally found that plasma levels of glutamine, carnitine and formate were changed significantly by the space flight (Supplementary Fig. 3). These results imply that Nrf2-deficiency itself evoked similar changes to those provoked by space stresses.

An intriguing hypothesis here was that these changes in metabolites recapitulated the changes in metabolites accompanied with ageing of humans. To address this hypothesis, we exploited human metabolome data accumulated in the Tohoku Medical Megabank (ToMMo) project[22]. We examined these metabolites in plasma from young age (20–40 years old) and old age (60–80 years old) participants in the population-based prospective cohort studies of ToMMo. We found that plasma glycerol level was increased in 60–80 age group (Fig. 4f), showing very good accord with the space mouse results. Plasma levels of glycine (Fig. 4g) and succinate (Fig. 4h) were lower in 60–80 age group than in 20–40 age group. These changes again showed very good agreement with those in the flight mice. Importantly, the latter two metabolites were also decreased in Nrf2-KO mice, supporting the notion that Nrf2 is important to decelerate the ageing of animals. In contrast, while plasma levels of glutamine, carnitine and formate were changed significantly by the space flight (Supplementary Fig. 3), the levels either changed moderately (glutamine and carnitine) or to a reversed-direction in the human ageing analysis (Supplementary Fig. 5). These results thus suggest that space stress induces ageing-like changes within a subset of plasma metabolites, and that Nrf2-deficiency also provokes similar ageing-like changes of plasma metabolites (Fig. 4i).

In addition, we also examined whether space flight induced ageing-like changes in gene expression. GSEA using the gene set of aged mice (Enrichr) revealed that space-induced changes of gene expression were enriched in ageing changes in liver, TpB, BAT, and WAT (Supplementary Fig. 6). These results demonstrate that space stress induced ageing-like changes in metabolites and gene expression.

**Lack of body-weight-gain in Nrf2-KO mice during space flight**. Upon return to Earth, we examined the overall health status of FL and GC mice, conducted behavioral examinations, and, after necropsy, histological examinations of multiple tissues of the mice. One of the most obvious phenotypes we found in these examinations was the lack of body-weight-gain specifically in Nrf2-KO mice during the space flight (Fig. 5a). Since we launched

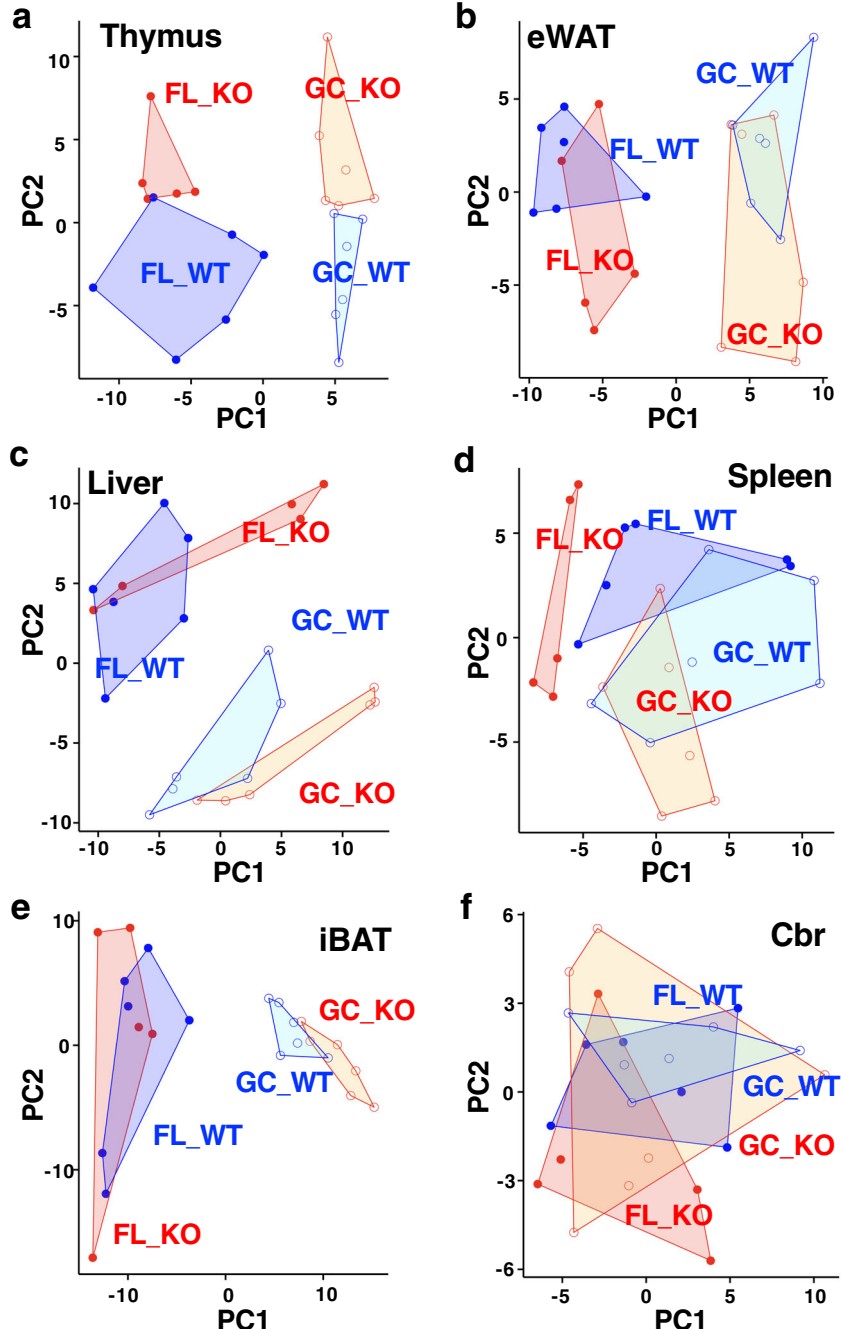

**Fig. 3 RNA-sequence analyses of flight and ground control mice. a–f** Plots for PCA applied to RNA-sequencing data of thymus (**a**), eWAT (**b**), liver (**c**), spleen (**d**), iBAT (**e**), or Cbr (**f**) from WT and Nrf2-KO mice in GC and FL groups. Dots represent individual animals. $n = 6$ for GC_WT, FL_WT and GC_KO, and $n = 5$ for FL_KO.

11-week-old mice, the mice were still gaining weight. In fact, FL_WT mice as well as both GC_WT and GC_KO mice gained body-weight almost to the same extent.

We then examined weights of organs and tissues of these mice, including eWAT, iBAT, liver, spleen, lung, thymus, heart, testis, and kidney (Fig. 5b–d and Supplementary Fig. 7). We found that eWAT weight, as percent of body weight, was significantly increased by space fight, but this increase was canceled largely in the FL_KO mice (Fig. 5b). Importantly, this decrease of eWAT weight did not occur in GC_KO mice. Similarly, weights of iBAT were significantly increased in FL_WT mice. However, in contrast to the situation of eWAT, the increase was not canceled in the FL_KO mice (Fig. 5c). Showing stark contrast to these two adipose tissues, the liver weight

proportional to body weight did not change much by either space flight or Nrf2-deficiency or both (Fig. 5d).

We designed an apparatus to monitor food-intake and water-consumption of mice while in space (Fig. 5e). Importantly, there were no significant differences in food-intake and water-consumption between WT and Nrf2-KO mice, both in-flight and on the ground (Fig. 5f, g). These results demonstrate that during space flight Nrf2-deficiency results in the reduction of body weight without changing food-intake and water-consumption. Available lines of evidence suggest that this may be linked to the diminished weight-gain of abdominal adipose tissues.

Therefore, to obtain further insight into the effects of space flight on lipid and glucose status in animal body, we analyzed the small

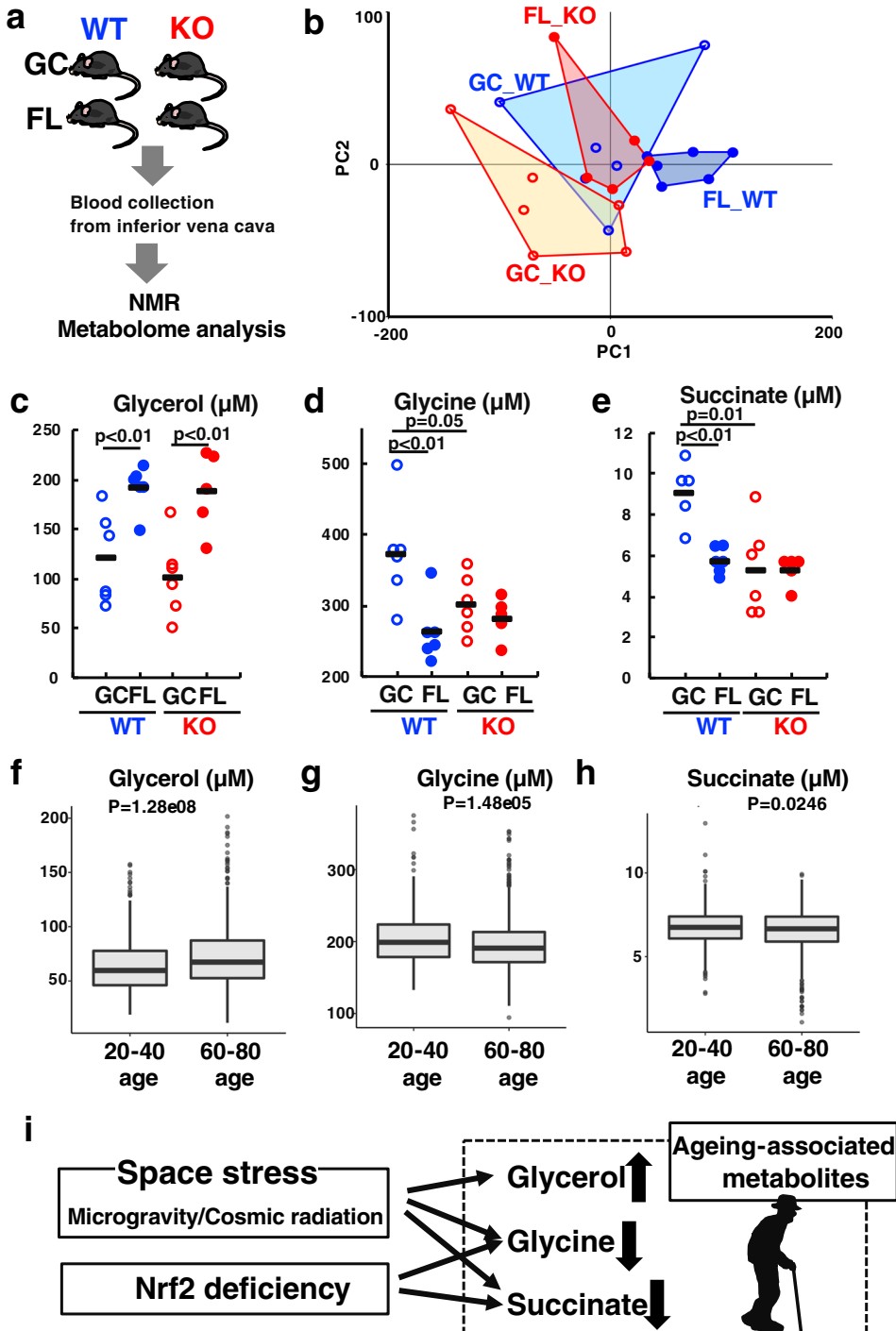

**Fig. 4 Space flight induces ageing-like changes of plasma metabolites. a** NMR-based metabolome analyses of plasma collected from the inferior vena cava of WT and Nrf2-KO mice in GC and FL groups. **b** Plots for PCA applied to NMR-based metabolome analyses of plasma from WT and Nrf2-KO mice in GC and FL groups. Dots represent individual animals. **c–e** Plasma levels of glycerol (**c**), glycine (**d**), and succinate (**e**) in WT and Nrf2-KO mice in GC and FL groups. Data are presented as mean, and dots represent individual animals. $n = 6$ for GC_WT, FL_WT and GC_KO, and $n = 5$ for FL_KO. One-way ANOVA with Tukey–Kramer test. **f–h** Age-distribution of plasma levels of glycerol (**f**), glycine (**g**), and succinate (**h**) in a human cohort of the ToMMo study (data replotted from ref. [22]). Center line, median; box limits, upper and lower quartiles; whiskers, 1.5× interquartile range; points, outliers. *P* values were calculated with Wilcoxon–Mann–Whitney test. $n = 545$ (20–40 age) and 955 (60–80 age) for (**f**, **g**). $n = 543$ (20–40 age) and 948 (60–80 age) for (**h**). **i** Identification of ageing-associated metabolites from space mouse experiment and the ToMMo human cohort study.

amounts of blood plasma collected from the mouse tail during flight (after 18 days launch, L + 18) and 2 days after return to Earth (R + 2). To our best knowledge, this is the first analysis of blood obtained from mice in space. We conducted mass-spectrometry metabolome analyses of the plasma (Fig. 5h). While there was no difference in

total cholesterol ester (CE) levels between WT and Nrf2-KO mice during flight (L + 18) (Fig. 5i), there was an elevation of total CE levels in FL_WT mice, but not Nrf2-KO mice after return to Earth (R + 2), where levels mirrored those of GC levels (Fig. 5j). While we do not have solid explanation for the increase of total CE level only

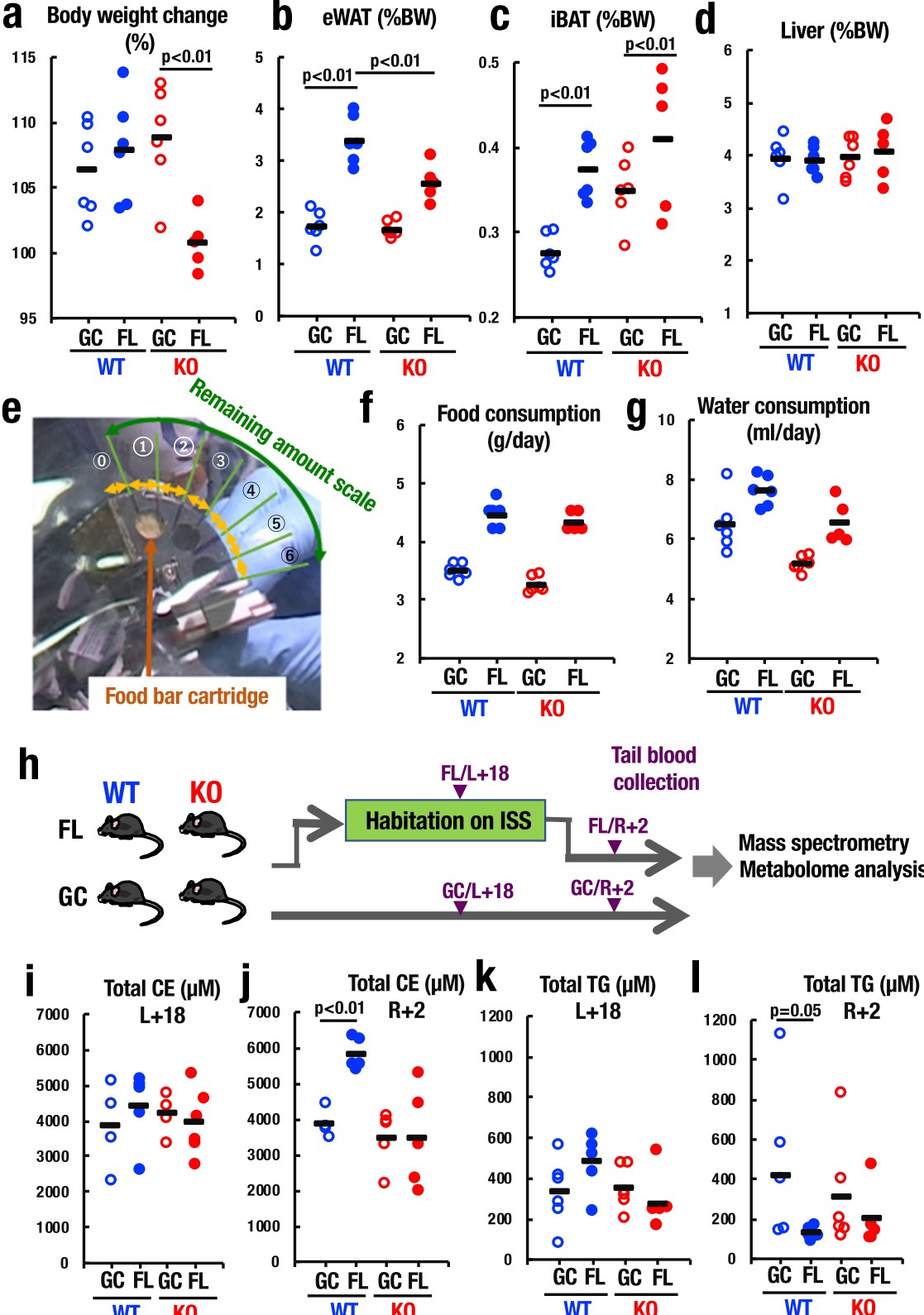

in the WT mice after return to Earth (R + 2), it plausible that the decrease of eWAT in FL_KO mice observed after return to Earth (R + 2) (Fig. 5b) might be associated with the unchanged total CE level in the plasma of these mice.

Similarly, total triglyceride (TG) levels did not change much between FL_WT and GC_WT mice, and also between FL_KO and GC_KO mice during flight (L + 18) (Fig. 5k). However, in contrast to the total CE level, the total TG level was significantly decreased in FL_WT mice upon return to Earth (R + 2) compared with GC_WT mice of R + 2 (Fig. 5l). This tendency was similar in GC and FL Nrf2-KO mice after return to Earth (R + 2) (Fig. 5l). Considering the elevated glycerol level in plasma (Fig. 4c), lipolysis from TG to glycerol was likely accelerated in flight mice in both WT and Nrf2-KO mice. In addition, our NMR metabolome analysis revealed that

**Fig. 5 Nrf2 deficiency affects body weight gain and abdominal white adipose tissue. a** Body weight change of WT and Nrf2-KO mice from before launch to after flight in GC and FL groups. Note that Nrf2-KO mice failed to gain body weight during space flight unlike WT mice. $n = 6$ for GC_WT, FL_WT and GC_KO, and $n = 5$ for FL_KO. **b–d** Body weight-normalized masses of eWAT (**b**), iBAT (**c**), and liver (**d**) from WT and Nrf2-KO mice in GC and FL groups. $n = 6$ for GC_WT, FL_WT and GC_KO, and $n = 5$ for FL_KO. Note that eWAT increased significantly in WT mice during space flight, but the increase was smaller in Nrf2-KO mice. **e** Food intake was estimated by remaining food scale of food bar cartridge. **f, g** Food intake (**f**) and water consumption (**g**) of WT and Nrf2-KO mice in GC and FL groups. $n = 6$ for GC_WT, FL_WT and GC_KO, and $n = 5$ for FL_KO. **h** Timeline for mass spectrometry-based metabolome analyses of plasma from tail blood of WT and Nrf2-KO mice in GC and FL groups. **i–l** Plasma levels of total cholesterol ester (CE) (**i, j**) and total triglyceride (TG) (**k, l**) in WT and Nrf2-KO mice in GC and FL groups during flight (L + 18) (**i, k**) and after return (R + 2) (**j, l**). Data are presented as mean, and dots represent individual animals. $n = 4$ (GC_WT), $n = 5$ (FL_WT), $n = 4$ (GC_KO), and $n = 5$ (FL_KO) for (**i**). $n = 5$ (GC_WT), $n = 5$ (FL_WT), $n = 5$ (GC_KO), and $n = 5$ (FL_KO) for (**j**). $n = 6$ (GC_WT), $n = 5$ (FL_WT), $n = 6$ (GC_KO), and $n = 5$ (FL_KO) for (**k**). $n = 5$ (GC_WT), $n = 6$ (FL_WT), $n = 6$ (GC_KO), and $n = 5$ (FL_KO) for (**l**). One-way ANOVA with Tukey–Kramer test.

there was no significant change in plasma glucose levels (Supplementary Fig. 5). These results thus demonstrate that the space travel affects lipid metabolism in mice, and the changes in certain aspects of lipid metabolism were either reversed or exacerbated by the loss-of-Nrf2 function.

**Nrf2 is critical for maintenance of WAT homeostasis in space.** We then extended our mouse analyses to the histology of abdominal WAT. We observed that lipid droplet size of eWAT in GC Nrf2-KO mice was significantly larger than that of GC_WT mice (Fig. 6a) despite noting that weights of eWAT were comparable between these two GC groups (Fig. 5b). Surprisingly, space flight gave rise to a marked increase of lipid droplet size in WT mice and to some extent in Nrf2-KO mice (Fig. 6a). We measured sizes of lipid droplets of all mice, and found that these changes during the space flight were quite reproducible (Fig. 6b). We also measured the distribution of lipid droplet sizes and confirmed that space flight provoked increases of lipid droplet size in both WT and Nrf2-KO mice (Fig. 6c, solid lines) compared with two groups of GC mice (dotted lines).

We calculated adipose cell number of eWAT utilizing these data, and found that the number of adipose cells in Nrf2-KO mouse eWAT was significantly lower than that in WT mouse eWAT (Fig. 6d), indicating that eWAT weight of Nrf2-KO mice on the ground is maintained by a complementary increase in droplet size. Intriguingly, lipid droplet size of FL_WT mouse eWAT became much larger following space flight than that of the GC mice. However, the lipid droplet size in eWAT of FL_KO mice was almost comparable with that of FL_WT mice (Fig. 6b), indicating that eWAT of the Nrf2-KO mouse did not become larger during space flight (Fig. 6e). Taken together, we propose that Nrf2 plays important roles in the maintenance of abdominal adipose tissue homeostasis, but that space stresses markedly affect this homeostasis regardless of the presence of Nrf2.

In an associated observation, lipid droplet size and thickness of subcutaneous fat was found to be increased by space flight, although there was no significant difference between WT and Nrf2-KO mice (Supplementary Fig. 8). Furthermore, numbers of hepatic Oil Red O-positive lipid droplets were increased by space flight (Supplementary Fig. 9). Tissue weights and lipid droplet sizes of iBAT also increased through space flight in both WT and Nrf2-KO mice (Fig. 5c and Supplementary Fig. 10). These wide-ranging observations demonstrate that adipose tissues in the whole body became larger in response to space stresses, supporting the contention that Nrf2 is critical for the maintenance of homeostasis of abdominal WAT.

**Space flight and Nrf2-KO induce metabolic impairment in eWAT.** To elucidate how space flight and Nrf2-deficiency induce perturbations in eWAT, we examined the transcriptome. Analyses of eWAT-derived RNA revealed that mRNAs coding for

genes involved in the respiratory chain (Fig. 7a) and fatty acid β-oxidation (Fig. 7b) were markedly reduced in eWAT from the flight mice of both genotypes compared with expression levels in eWAT from GC WT mice. In addition, similar but milder changes than those in the flight mice were observed in the GC_KO mice (Fig. 7a, b), indicating that space stresses induce metabolic impairment in eWAT, which is qualitatively similar, but much more severe, than those observed with Nrf2-deficiency mice on Earth. These results suggest that these reductions of mitochondrial activity might result in the enlargement of adipocytes in FL_WT and GC_KO mice.

Further transcriptome analysis revealed that expression levels of diabetes-related chemokine genes[23] were increased in eWAT from flight mice of both genotypes compared to GC mice (Fig. 7c). Consistent with this observation, expression levels of marker genes for macrophage (*Cd68*, *Lgals3*, and *Adgre1*) and angiogenesis (*Kdr* and *Pecam1*) were increased in eWAT from flight mice of both genotypes compared to GC mice (Fig. 7d). Since angiogenesis sustains inflammation by delivering oxygen and nutrients for inflammatory cells, and inflammation in turn can cause insulin resistance[24], these results suggest that space stresses might induce adipose tissue growth by means of inflammation and angiogenesis.

In order to gain insight as to how Nrf2 deficiency on Earth leads to the decrease in adipocyte number, we searched for changes in gene expression in eWAT from GC_KO mice. We found that expression levels of many PPARγ-target genes[25] were down-regulated in eWAT from GC_KO mice compared to GC_WT mice (Fig. 7e). Since PPARγ is important for adipocyte differentiation[25], this downregulation of PPARγ in GC_KO mice might affect adipocyte differentiation, provoke reduction of adipocyte numbers and, in turn, give rise to compensatory enlargement of the adipocytes, thereby sustaining adipose mass of eWAT on Earth (Fig. 7f). The transcriptome analysis also revealed that expression of a number of PPARγ-target genes were changed in the eWAT of flight mice, showing similar profiles irrespective of the Nrf2 genotypes (Fig. 7e). The profile of flight mice showed remarkable differences from those of Nrf2 KO mice on Earth. These results thus indicate that space stresses elicit much stronger influences on the PPARγ-target gene expression, along with the respiratory chain, fatty acid β-oxidation, chemokine and macrophage-/angiogenesis-related gene expressions, than the Nrf2 knockout does. We surmise that these changes in the gene expression profiles are, at least in part, responsible for the marked enlargement of the adipose tissues during the space travel.

**Discussion**

Nrf2 is the key regulator of the adaptive response against various environmental and endogenous stresses[13]. Since oxidative stress activates the Nrf2 signaling pathway through stress-sensing mediated by the Nrf2 chaperone Keap1 (Kelch-like ECH-associated protein 1)[26], cosmic radiation and/or microgravity during space travel may well activate Nrf2 through the generation

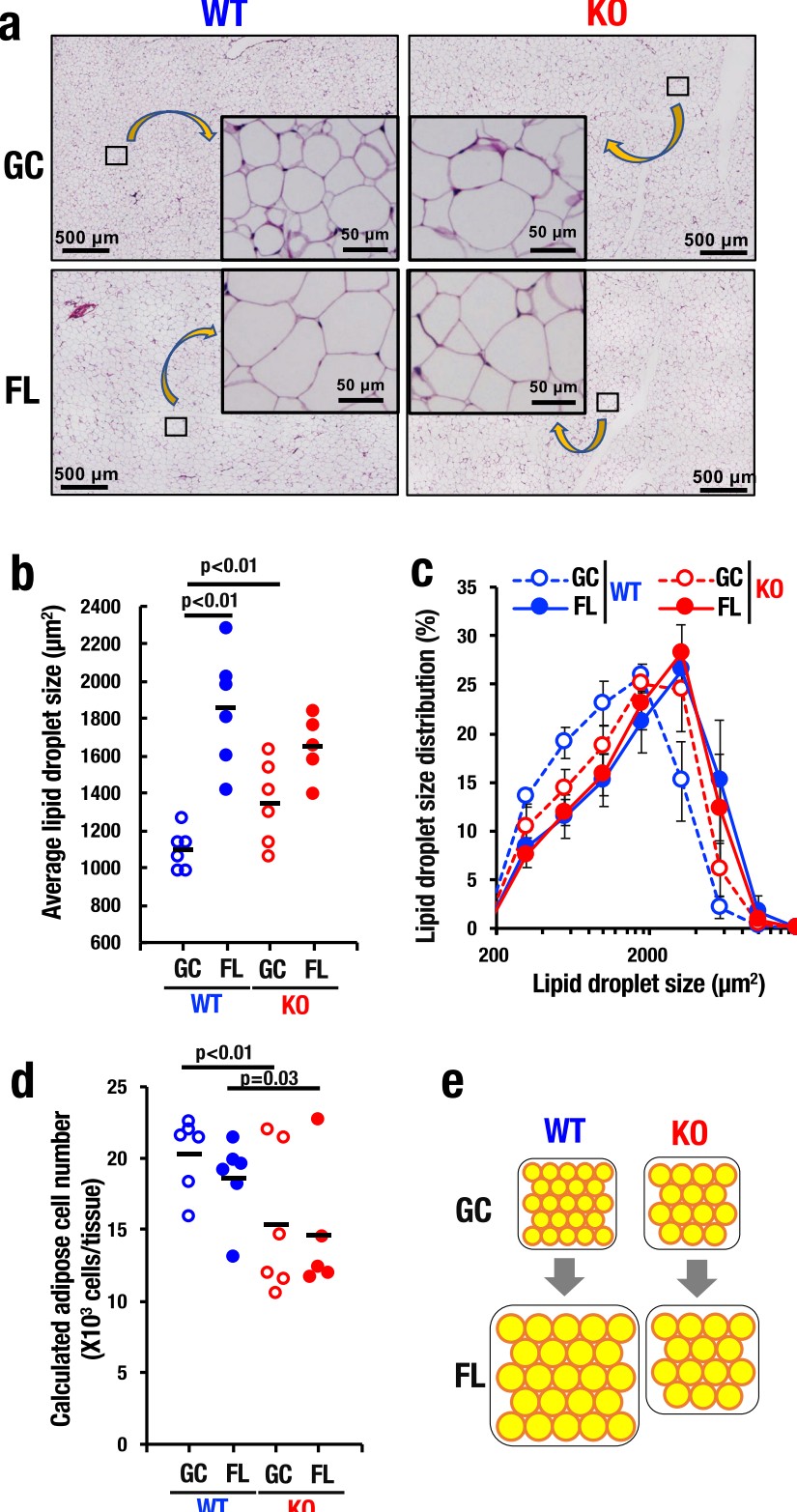

**Fig. 6 Nrf2 is essential for maintaining eWAT mass in space. a** Histological images of eWAT from WT and Nrf2-KO mice in GC and FL groups. **b**, **c** Average lipid droplet size (**b**) and lipid droplet size distributions (**c**) of eWAT from WT and Nrf2-KO mice in GC and FL groups. Data are presented as mean, and dots represent individual animals. $n = 6$ for GC_WT, FL_WT and GC_KO, and $n = 5$ for FL_KO. **d** Calculated adipose cell number of eWAT from WT and Nrf2-KO mice in GC and FL groups. Data are presented as mean, and dots represent individual animals. $n = 6$ for GC_WT, FL_WT and GC_KO, and $n = 5$ for FL_KO. **e** Model for adipose cells in eWAT from WT and Nrf2-KO mice in GC and FL groups. Data are presented as mean, and dots represent individual animals. One-way ANOVA with Tukey–Kramer test.

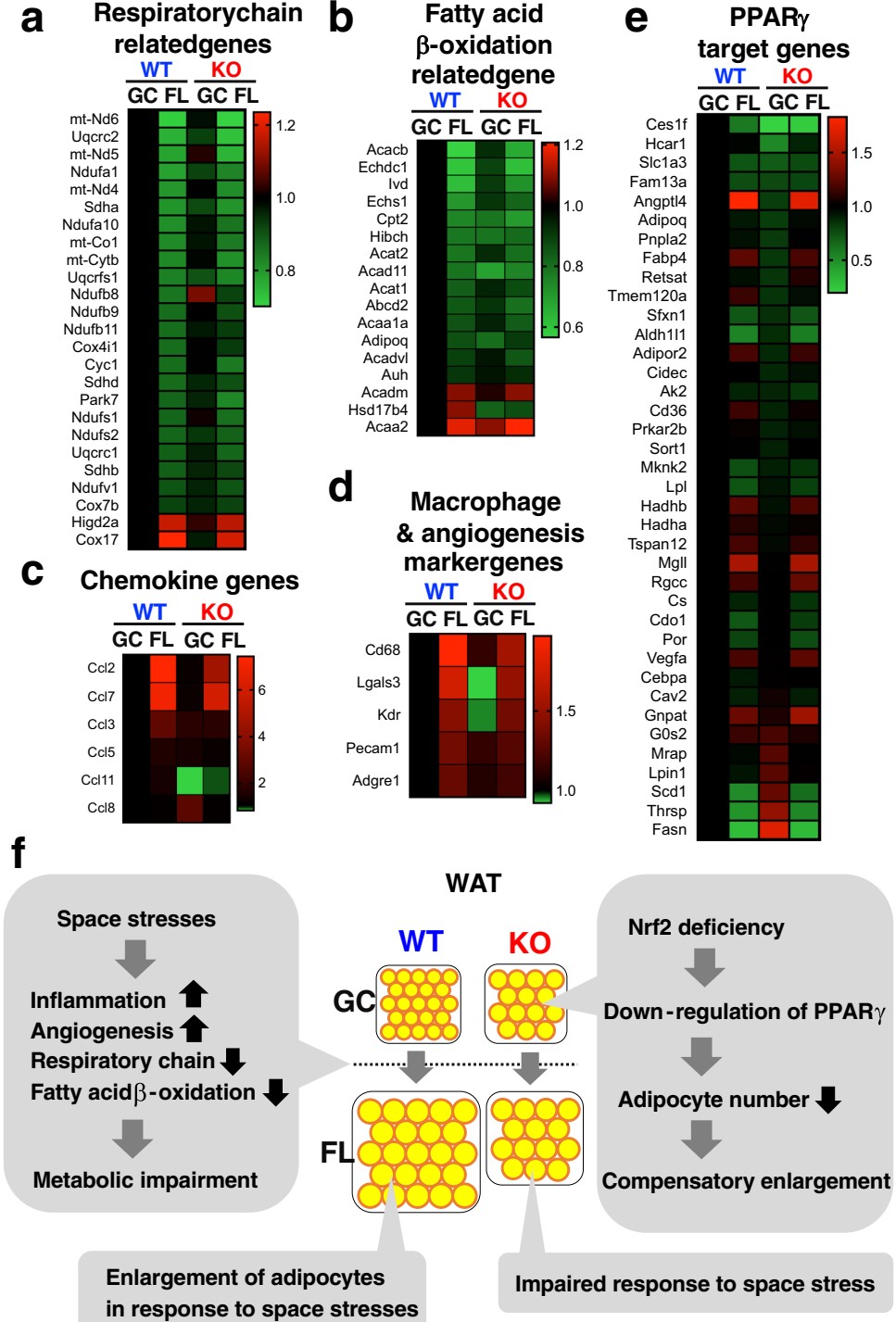

**Fig. 7 Space stresses and Nrf2-deficiency induce metabolic impairment in eWAT. a–e** Heatmap of expression levels of genes involving in the respiratory chain (**a**), fatty acid β-oxidation (**b**), chemokine (**c**), macrophage and angiogenesis (**d**) and downstream of PPARγ (**e**) in eWAT from WT and Nrf2-KO mice in GC and FL groups. **f** Hypothetical model for space-induced metabolic impairment in eWAT from WT and Nrf2-KO mice in GC and FL groups.

of oxidative stress. Similarly, there is a possibility that mechanical stress caused by microgravity may contribute to Nrf2 activation during space travel. However, there has been limited direct examination as to whether these space stresses activate the Nrf2 signaling pathway and to what extent activation of the pathway may be protective against these stresses. Therefore, we conducted a space travel experiment utilizing Nrf2 knockout mice. To our best knowledge, this is the first attempt of space travel for gene-knockout disease model mice in which mice are

returned safely to Earth. Comprehensive RNA sequencing analyses of various organs/tissues from the returned mice revealed that space stresses indeed have induced the Nrf2 signaling pathway in many tissues. Our metabolome analysis further revealed that the stresses during the space travel have induced ageing-related changes of some plasma metabolites. Another salient finding in the space travel of Nrf2 knockout mice is that Nrf2 is important for maintaining homeostasis of white adipose tissues. Collectively, these results thus unequivocally demonstrate

the importance of the Nrf2 signaling pathway in responses to the stresses of space travel.

Many technological advances contributed to the success of this space mission, including the mouse transport and habitat systems. Of direct application to the biomedical inquiries, we developed a new blood collection procedure that is minimally invasive and easy-for-use for astronauts. This apparatus is important since there is a limitation for the training of astronauts for specialized operations such as blood collection. Capitalizing on the development of the new device, we successfully collected small amounts (40 μL) of blood and plasma from mouse tails during the flight. Subsequent mass spectrometer-based metabolome analysis successfully detected plasma metabolites in the small volumes of plasma. This newly established procedure of blood collection from mice during space travel coupled with a highly sensitive metabolome analysis brings us a powerful approach to elucidate physiological and pathological changes of intermediary metabolism in gene-modified or other model mice during space travel.

Indeed, the systematic metabolome analysis in this study revealed that space flight induced increases in glycerol and decreases in TG, implicating the occurrence of enhanced lipolysis in mice during the flight. It has been shown that lipolysis is stimulated by stress-induced catecholamines, including adrenaline and noradrenaline[27], and an increase in circulating levels of catecholamines has been observed commonly in astronauts[28]. Therefore, we surmise that the enhanced lipolysis is the consequence of elevated catecholamine hormones. Alternatively, the enhanced lipolysis in mice might be a compensatory response to the increase in adipose mass occurring during space travel. In this regard, it is interesting to note that the increase of plasma glycerol shows significant association with human ageing as observed in the ToMMo cohort study. Now, verification of this relationship in prospective cohort studies becomes quite important and intriguing, since association of the glycerol increase and ageing has not been recognized heretofore.

We also observed that decreased plasma glycine levels showed a marked association with ageing. In contrast to the situation for glycerol, the association of plasma glycine levels with ageing has been reported, such that dietary glycine supplementation extended the lifespan of rats[29] and addition of glycine to the culture media restored the phenotype of aged cells back to that of young cells[30]. Taken together, our results and these reports indicate that space travel induces in mice an ageing phenotype associated with these plasma metabolites.

Our finding that Nrf2 disruption impairs body-weight gain and WAT homeostasis in space mice during flight elicits several important considerations. First, it has been known that there is a specific regulatory single nucleotide polymorphism (rSNP) in the *Nrf2* gene that downregulates the expression of Nrf2 and increases the risk of a few diseases in humans[31] and mice[32]. Thus, a number of questions arise related to this rSNP. For instance, does presence of this rSNP in the *NRF2* gene influence the health of future longer-term space travelers? Do minor allele homozygote mice of the rSNP retain exacerbated risk for body-weight reduction and/or perturbation of WAT homeostasis by long-term space flight or by ageing? Second, small molecule inducers of the Nrf2 signaling have been developed or are under development[33,34]. These NRF2 inducers can be taken as drugs or through foods and dietary supplements. We surmise that these NRF2 inducers may serve to mitigate some of the stresses associated with space travel.

We have obtained substantive amounts of informative data on this MHU-3 space–flight study. However, we could analyze only limited sets of data in a timely manner. In fact, we feel that many changes were induced in a space–flight specific manner and/or Nrf2-knockout mouse specific manner that have not been recognized through our first-pass analyses. We therefore present our series of histological analyses as Supplementary Data (Supplementary Figs. 11–14). Further metabolomic analyses, blood analyses, kidney function analyses, behavioral analyses, and muscle analyses, will be published separately. Collectively, this study has pioneered the future of experiments utilizing gene-modified model mouse defective in adaptive responses to stresses associated with space travel. We believe that continuation of this series of space mouse studies will provide insightful information useful to overcome possible mission-limiting, space flight-derived stresses evoked by human space exploration.

## Methods

**The MHU-3 project**. The HCU rev.1 is an onboard habitation cage that accommodates one mouse per cage[11,12]. The HCU is equipped with a food bar, watering system (two redundant water nozzles and a water balloon acting as power-free pressure source), an odor filter, two fans (for redundancy) for air ventilation, waste collecting equipment, an LED/IR video camera with a wiper inside the cage to keep the observation window clean. The health of each mouse was determined based on the conditions of eyes, ears, fur, and tail as observed in the transmitted videos. Paper sheets were mounted on the cage wall to quickly eliminate liquid, such as urine, from the cage. A photocatalytic thermal spray was applied to the sheets for deodorant and antibacterial effects. Air ventilation inside the cage was maintained by airflow (<0.2 m/s) generated by fans on the HCU. Differences in the volume of air ventilation and airflow rates between HCU in GC and microgravity conditions were negligible because the fans regulating ventilation in both gravity conditions were maintained at the same speed. The day/night cycle used 12-h intervals. Food and water were replenished once a week. Temperature, humidity, carbon dioxide, and ammonia were monitored precisely and recorded in logs.

The TCU was used to transport mice aboard the SpaceX Dragon capsule during the launch and return phases, and was placed in a powered locker sized for ISS single cargo transfer bag. The TCU contains 12 cylindrical cages for housing mice individually. The TCU is equipped with a cylindrical food bar, watering system (a water nozzle and two water balloons), an odor filter, two fans (for redundancy), waste collecting equipment, and LEDs with lights for day/night cycle. Paper sheets mounted in the waste collection area were treated with photocatalytic thermal spray for deodorant and antibacterial effects, similar to the HCU. A temperature/humidity logger was attached on the TCU air inlet to monitor the environment during launch and reentry operations.

**Animals**. Nrf2-KO (Nfe212tm1Ymk)[16] and WT male mice in the C57BL/6J background were bred at Charles River Laboratories Japan for MHU-3. All animal experiments were approved by the Institutional Animal Care and Use Committees of JAXA (protocol numbers 017-001 and 017-014), NASA (protocol number FLT-17-112), and Explora BioLabs (EB15-010C), and conducted according to the related guidelines and applicable laws of Japan and the United States of America.

**Pre-launch acclimation activities and animal selection**. Three weeks prior to launch, 60 WT and 60 Nrf2-KO mice (8 weeks old) were delivered from Charles River Laboratories Japan to KSC. These mice were acclimatized to the environment in individual housing cages (Small Mouse Isolator 10027, Lab Products) in an air-conditioned room (temperature: 23 ± 3 °C; humidity: 40–65%) with a 12:12-h light/dark cycle at the SSPF Science Annex at KSC. Three acclimation phases were set: Phase I, body weight recovery phase; Phase II, water nozzle acclimation phase and Phase III, flight food acclimation phase. After approximately three weeks of acclimation, the mice were ready for transport to the ISS and had reached the age of 12 weeks. Succinctly, at Phase I, mice were fed CRF-1 and given autoclaved tap water ad libitum using ball-type water nozzles. The bedding material consisted of paper chips (ALPHA-DRI). At Phase II, water nozzles are changed to flight nozzles. Finally, at Phase III, the food was changed to flight food. Fecal collection and body swabs were obtained, and these samples were sent to Charles River Laboratories in USA for pol-based SPF testing.

As some of the Nrf2-KO mice harbor an intrahepatic shunt[35], Nrf2-KO mice were challenged with 100 mg/kg ketamine and 5 mg/kg xylazine and recovery times were monitored at the age of 9 weeks to eliminate the mice harboring an intrahepatic shunt. Nrf2-KO mice with long sleeping duration were judged as mice without shunts. Blood was taken from tails of all 120 mice.

During the acclimation phase, body weight and food/water consumption were measured and recorded. We selected 12 mice for flight and backup, respectively. The selection of flight candidate mice was based on body weight, food consumption, water intake, and absence of hepatic shunt. We also checked the health of each flight candidate mouse by evaluating the condition of their eyes, ears, teeth, fur, and tail. The five subgroups of flight candidates were necessary to support possible launch attempts in case of unexpected postponements.

**Onboard operations by ISS crew**. One day prior to launch, 12 mice were loaded into the TCU and made ready for launch. Mice were transported to the ISS by SpX14. After the Dragon vehicle of SpX14 berthed with the ISS, mice were

transferred to the HCU by a crew member. Mouse-husbandry tasks included exchanging food cartridges, supplying water, collecting waste, and replacing odor filters. The block food (approximately 35 g) is contained in the cartridge, which supports 1-week habitation. The cartridges have windows and scale at their sides, therefore the remaining amount of the foods was estimated, via video without returning them to the ground, at weekly food cartridge exchange operations. After 31 days onboard the ISS, mice were transferred to the TCU for the return to Earth.

**Return phase and animal dissection**. After unberthing from the ISS, the Dragon vehicle splashed down in the Pacific Ocean off the coast of California. After a ship picked up the cargo, the returned TCU was transported to a port in Long Beach. JAXA received the TCU from NASA and transported them to Explora BioLabs in San Diego using an environmentally controlled van. After the health and body weights of mice were assessed, open field, light/dark transition and Y-maze tests were performed. Blood was collected from tail after the behavioral tests. Isoflurane-anesthetized mice were then euthanized by exsanguination and dissected for the collection of tissue samples.

**GC experiment**. A GC experiment that simulated the space experiment was conducted at JAXA Tsukuba in Japan from September 17 to October 20, 2018. Six WT and six Nrf2-KO mice without the hepatic shunts were individually housed in the same way as for the flight experiment. Both the TCU and HCU were placed in an air-conditioned room (average temperature: 22.9 °C; average humidity: 48.4%) with a 12:12-h light/dark cycle. Fan-generated airflow (0.2 m/s) inside the HCU maintained the same conditions as in the flight experiment. The mice were fed CRF-1. The bedding material consisted of paper chips (ALPHA-DRI) in the acclimation cages, but was not used in the HCU. Feeder cartridges and water bottles were replaced once a week, and cages in the HCU were not replenished. After the health and body weights of mice were assessed, open field, light/dark transition and Y-maze tests were performed. Blood sampling from the tail was conducted as for the flight experiment. All mice were anesthetized by isoflurane inhalation and euthanized under anesthesia, and then dissected to collect tissue samples.

**Blood collection procedure**. During the flight experiment, blood samples were collected from the distal end of the tail with tail clippers (KAI, PQ3357). The tail clippers were mounted on a guard plate adjusted to 1 mm from the tip of the clippers, and all such tail clippers used in this experiment were inspected to confirm each tool's ability to amputate less than 1 mm of the tail prior to launch.

Each mouse was transferred from the HCU to the restrainer (Sanplatec, 99966-31), which was pre-warmed by disposable heat pads (Kiribai Chemical; New Hand Warmer Mini) to increase obtainable blood volume. The inner surface temperature of the restrainer was approximately 45 °C. After being disinfected, the tail was milked after being clipped by a tail clipper to collect a blood sample into capillary tubes (Drummond Scientific; 8-000-7520-H/5). To promote hemostasis, the tip of mouse tail was compressed for one minute with a hemostat (Ethicon, 15726). The blood samples collected in capillary tubes were centrifuged and immediately frozen in a −80 °C-freezer without being snap-frozen. Prior to blood collection, a veterinarian checked the health status of all mice via videos and confirmed their adaptation to the space environment.

Before and after the flight experiment, blood samples were taken by nicking the tail. Mice were transferred to the restrainers and then nicked using disposable scalpels (FEATHER Safety Razor, No. 10). After the nicking or the on-orbit procedure, blood was collected in the capillary tubes and centrifuging was conducted. The procedures for both nicking and clipping were in accordance with NIH guidelines.

**Micro-computed tomography (microCT) analysis**. The right femurs of mice were fixed in 70% ethanol and distal regions were analyzed as described[11]. MicroCT scanning was performed using a ScanXmate-A100S Scanner (Comscantechno). Three-dimensional microstructural image data was reconstructed and BMD was calculated using TRI/3D-BON software (RATOC System Engineering) in accordance with the guidelines.

**RNA-sequence analysis**. Total RNA was isolated from temporal bone, mandibular bone, spleen, liver, white adipose tissue, brown adipose tissue, cerebrum, kidney, and thymus. RNA integrity was assessed using an Agilent 2200 TapeStation. For each tissue, 1.0 μg of total RNA was used for further steps. Total RNA samples were subjected to isolation of poly(A)-tailed RNA and library construction using Sureselect Strand Specific RNA Sample Prep Kit (Agilent Technologies), except that total RNA from thymus was subjected to ribosomal RNA depletion using Ribo-Zero rRNA Removal Kit (Illumina) and library construction using the similar step. The libraries were sequenced using HiSeq2500 (Illumina) for 76 cycles of single read and more than 17 million reads were generated per sample. The raw reads were mapped to the mouse mm10 genome using STAR (version 2.6.1)[36]. Transcripts per million (TPM) values were obtained to measure gene expression using RSEM (version 1.3.1)[37]. The TPM was normalized in the Subio Platform software (Subio). Genes with mean TPM less than five among all groups were excluded. The PCA was performed using the prcomp function in R software (version 3.5.1; www.r-project.org). R-based heatmap.2 in gplots package was used for generating the heatmap. The differential gene expression analysis was performed using DESeq2 (version 1.22.2)[38]. GSEA[39] (www.broad.mit.edu/gsea)

was used to assess space flight-induced changes of gene expression. The GSEA was performed using previously published ageing signatures in Enrichr[40] (GSE20425 for liver, GDS3028 for temporal bone, GSE25325 for BAT and GSE25905 for WAT) or 50 hallmarked gene sets of MSigDB v7.1 (www.gsea-msigdb.org/gsea/msigdb/index.jsp).

**Histological analysis**. Tissues were fixed in Mildform 10N (Wako Pure Chemical) and processed into paraffin-embedded tissue sections. The sections were stained with hematoxylin and eosin. Lipid droplet size was measured by BZ-X800 (Keyence), and adipose cell number was calculated from their tissue weights and droplet sizes[41,42]. To visualize hepatic lipid content, livers were fixed with 4% paraformaldehyde and embedded in OCT (Tissue Tek). The frozen sections were stained with Oil Red O (Sigma Aldrich) and counterstained with hematoxylin.

**Plasma analysis by NMR spectroscopy**. Plasma metabolites of the blood samples obtained from inferior vena cava were analyzed using NMR spectroscopy[17,18]. Plasma metabolites were firstly extracted using a standard methanol extraction procedure using 50 μL of plasma per sample. The supernatant was transferred to a new tube and evaporated. Each dried sample was suspended in a 200-μL solution of 100-mM sodium phosphate buffer (pH 7.4) in 100% $D_2O$ containing 200-μM d6-DSS. NMR experiments were performed at 298 K on a Bruker Avance 600-MHz spectrometer equipped with a CryoProbe and a SampleJet sample changer. Standard 1D NOESY and CPMG (Carr-Purcell-Meiboom-Gill) spectra were obtained for each plasma sample. All data were processed using the Chenomx NMR Suite 8.3 processor module (Chenomx). Metabolites were identified and quantified using the target profiling approach implemented in the Chenomx Profiler module. The concentration of metabolites was analyzed by a PCA on SIMCA13.0.0 (Umetrics).

**Plasma analysis by mass spectrometry**. The plasma from tail blood sample was collected from the capillary tube and stored at −80 °C until analysis. Plasma (4 μL per analysis) were prepared using the protocol for the Absolute IDQ® p400 HR Kit (Kit400), which includes a detailed standard operating procedure (SOP) protocol with documentation for sample preparation, instrument setup, system suitability testing, and data analysis. The Kit400 quantifies 408 metabolites, and includes calibration standards, internal standards, and quality control samples. The ultra-high-performance liquid chromatography (UHPLC) system consisted of an online degasser, auto sampler, dual pump, and column oven (UltiMate™ 3000 RSLC system, Thermo Fisher Scientific), and a quadrupole Fourier transform mass spectrometry (FTMS, Q Exactive Orbitrap system). The operating conditions of the UHPLC-FTMS system and the details of data analysis followed the protocol of the Kit400 analysis[43].

**Statistical and reproducibility**. Data points represent biological replicates. Comparisons between groups were conducted using one-way ANOVA with Tukey–Kramer test or Wilcoxon–Mann–Whitney test. Data were considered statistically significant at $p < 0.05$.

**Reporting summary**. Further information on research design is available in the Nature Research Reporting Summary linked to this article.

## Data availability

The data discussed in this publication have been deposited in NCBI's Gene Expression Omnibus[44] and are accessible through GEO Series accession number GSE152382 (https://www.ncbi.nlm.nih.gov/geo/query/acc.cgi?acc=GSE152382). All relevant data are available from the corresponding authors upon reasonable request. Supplementary Data 1 contains gene set enrichment analysis (GSEA) of the gene expression changes shown during the space flight in wild-type mice. Source data for graphs and charts presented in the main figures are provided in Supplementary Data 2.

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

## Acknowledgements

We would like to thank Norishige Kanai (astronaut) for the onboard operation, and Toshiaki Kokubo and Noriko Kajiwara (JAXA visiting veterinarians) for monitoring mouse health. We also thank Naoko Ota-Murakami, Fumika Yamaguchi, Masumi Umehara, and the members of the mouse health check team, for performing daily onboard health checks, Ramona Bober, Autumn L. Cdebaca, Rebecca A. Smith for animal care and ground experiment supports, Hirochika Murase, Hiroaki Kodama, Yusuke Hagiwara, and members of hardware development team for MHU hardware preparation and operations, Kohei Hirakawa, Teruhiro Senkoji, Haruna Tanii, Motoki Tada, Yuki Watanabe, Kayoko Lino, Hiromi Sano, Yui Nakata, Hiromi Suzuki-Hashizume, Eiji Ohta, Osamu Funatsu, Hideaki Hotta, Hatsumi Ishida, Mariko Shimizu, and members of JEM operational team for the research coordination, Takahashi Ueda and Tomohiro Tamari for animal preparations, Hong Xin and Grishma Acharya for landing site operational supports, and Sayaka Umemura, Laura Lewis, Charles E. Hopper, Jennifer J. Scott Williams, Robert Kuczajda for international coordination. This work was selected as a space rodent research study for JAXA's feasibility experiments using ISS/Kibo announced in 2015, and also supported in part by MEXT/JSPS KAKENHI (19H05649 to M.Y. and 17KK0183, 18H04963, 19K07340, to T. Suzuki), Takeda Science Foundation (M.Y., and T. Suzuki.) and the Smart Aging Research Center, Tohoku University (M.Y.). This work was also supported in part by the grants JP19km0105001, JP19km0105002 and SHARE.

## Author contributions

T.S., A.U., A.Y., T.W.K., D.S., and M.Y. wrote the paper. A.Y., M. Shimomura, H. Mizuno, M. Shirakawa, and D.S. conducted the space experiments. T.S., A.U., A.Y., K.T., M. Suzuki, N.H., R.R., E.N., N.O., A.G., H.S., R.B., A.O., F.K., T.Y., D.S., S.K., T.N., S.F., H.I., K.N., N.S., I.H., R.S., T.O., H. Motohashi, H.T., R.O., T.K., and S.T. conducted the molecular, histological, physiological, and behavioral assessments and analyzed the data.

## Competing interests

The authors declare no competing interests.
