## [Peer Review File · Communications Biology]

Reviewers' comments:

Reviewer #1 (Remarks to the Author):

In their manuscript entitled 'Space Travel of Mice Demonstrates Contribution of Nrf2 to Maintenance of Homeostasis', Suzuki and colleagues study how the knockout of NRF2, a transcription factor that coordinates cellular stress responses, affects the adaptation of mice to stress induced by space flight. Whilst the aim of the current work (i.e. to determine the effect of Nrf2 deletion on the health of mice undergoing space travel) is rather vague, the study is original. The reported effects of space travel and Nrf2 deficiency on white adipose tissue homeostasis are very interesting and the main conclusions are supported by the data. In addition, the study reports important technological advances related with mouse housing and handling in space that contributed to the success of this space mission when compared to previous missions. These include the development of a minimally invasive blood collection procedure that allowed astronauts to collect small amounts of blood that were nonetheless compatible with mass spectrometer based metabolome analysis. I have no major issues related with the publication of this study.

Minor issues

P7, "The expression of these typical Nrf2 target genes were upregulated" should read "The expression of these typical Nrf2 target genes was upregulated".

P9, "this pattern reflected the cell heterogeneity suppressed of Cbr" should read "this pattern reflected the cell heterogeneity of Cbr".

P10, please re-write the following sentences, which appears to contain a repetition: "closer inspection indicated that PC1 separated FL Nrf2-KO vs. FL WT and GC Nrf2-KO vs. GC WT (Fig. 4b). Similarly, the PC1 also separated GC Nrf2- KO vs. GC WT and FL Nrf2-KO vs. FL WT." In fact, PC1 does not appear to separate GC Nrf2- KO vs. GC WT.

In the legend of Supplementary Video 2, please indicate mouse genotype and which mice returned from space or GC.

Reviewer #2 (Remarks to the Author):

This study examined multiple metabolic phenotypes that were affected by space flight in mice and the contribution of Nrf2 in these phenotypes. They demonstrated that Nrf2 was activated by space flight and that Nrf2 was required for maintaining WAT homeostasis during space flight. Understanding the role of Nrf2 in metabolic adaptation to space flight is interesting and important, but the major concern is that some conclusions in the paper were not fully supported by the data.

Major concerns:

1. Fig 2b showed that many of the Nrf2 targets were induced in FL-WT group compared to GC_WT. However, Fig 2C showed that both Nqo1 and Gstm1 were still induced by space flight in Nrf2 KO groups, comparing FL_KO with GC_KO. It suggests that space flight can induce these genes in a Nrf2 independent manner. The authors could analyze how many Nrf2 targets, which are induced by space flight in WT, are not induced by space flight in KO mice by comparing FL_KO with GC_KO.
2. The statement that space flight induces ageing-like changes of plasma metabolites is not supported by data. Only three metabolites that were similarly affected by space flight and ageing are not enough. The authors should explore more metabolites. A comprehensive comparison of metabolites that are affected by space flight and ageing would be important.
3. Does space flight also induce ageing-like changes of gene expression? The author can compare their RNA-seq data with some published ageing transcriptome data in mice.
4. Besides the Nrf2 targets and some metabolic genes, what genes are affected by space flight in

RNA-seq analysis? How many of them are dependent on Nrf2? Functional classification of these genes (such as GO term analysis) is also important. This will show the significance of Nrf2 during space flight adaptation.

To Reviewer #1:

The reported effects of space travel and Nrf2 deficiency on white adipose tissue homeostasis are very interesting and the main conclusions are supported by the data. In addition, the study reports important technological advances related with mouse housing and handling in space that contributed to the success of this space mission when compared to previous missions. These include the development of a minimally invasive blood collection procedure that allowed astronauts to collect small amounts of blood that were nonetheless compatible with mass spectrometer based metabolome analysis. I have no major issues related with the publication of this study.

We thank the reviewer for the professional comments.

Minor issues

P7, “The expression of these typical Nrf2 target genes were upregulated” should read “The expression of these typical Nrf2 target genes was upregulated”.

P9, “this pattern reflected the cell heterogeneity suppressedof Cbr” should read “this pattern reflected the cell heterogeneity of Cbr”.

We apologize for our oversights. We have corrected these grammatical errors.

P10, please re-write the following sentences, which appears to contain a repetition: “closer inspection indicated that PC1 separated FL Nrf2-KO vs. FL WT and GC Nrf2-KO vs. GC WT (Fig. 4b). Similarly, the PC1 also separated GC Nrf2- KO vs. GC WT and FL Nrf2-KO vs. FL WT.” In fact, PC1 does not appear to separate GC Nrf2- KO vs. GC WT.

We apologize for our carelessness. We have corrected the sentence to “closer inspection indicated that PC1 separated FL Nrf2-KO vs. FL WT (Fig. 4b)”.

In the legend of Supplementary Video 2, please indicate mouse genotype and which mice returned from space or GC.

We thank the reviewer for the comment. We have indicated genotypes and flight status of mice in the legend of Supplementary Video 2.

To Reviewer #2:

This study examined multiple metabolic phenotypes that were affected by space flight in mice and the contribution of Nrf2 in these phenotypes. They demonstrated that Nrf2 was activated by space flight and that Nrf2 was required for maintaining WAT homeostasis during space flight. Understanding the role of Nrf2 in metabolic adaptation to space flight is interesting and important, but the major concern is that some conclusions in the paper were not fully supported by the data.

We thank the reviewer for the professional and constructive comments. We agree with the comments and have answered the comments exploiting and showing details of the data obtained.

Major concerns:

1. Fig 2b showed that many of the Nrf2 targets were induced in FL-WT group compared to GC-WT. However, Fig 2C showed that both Nqo1 and Gstm1 were still induced by space flight in Nrf2 KO groups, comparing FL_KO with GC_KO. It suggests that space flight can induce these genes in a Nrf2 independent manner. The authors could analyze how many Nrf2 targets, which are induced by space flight in WT, are not induced by space flight in KO mice by comparing FL_KO with GC_KO.

We thank the reviewer for this insightful comment and we have reexamined our data to answer this comment. To address this point, we examined Nrf2-dependent and Nrf2-independent gene expressions by comparing the gene expressions of FL_KO mice with GC_KO mice, especially focusing on those of space flight-induced genes in WT mice. We are happy to report the results of this analysis.

As shown in Supplementary Figure 1, we found that space-induced changes of gene expression were classified into Nrf2-dependent group and Nrf2-independent group in various tissues. It is noteworthy that Nrf2-dependent space-induced genes include typical Nrf2 target genes. Based on these analyses, we replaced *Gstm1* (thymus) data with *Hmox1* (WAT) data in Fig. 2c, and added sentences explaining this observation to text (Page 8, line 1-7, 15-16).

As the reviewer pointed out, there must be Nrf2-independent but space-stress inducible changes in addition to the Nrf2-dependent changes, and these two forms of regulation appear to co-exist even in the expression of a single gene. We agree with this point, but we believe that the finding that there is a broad set of Nrf2-dependent changes of gene expression provoked by space stress is supported by these analyses.

2. The statement that space flight induces ageing-like changes of plasma metabolites is not supported by data. Only three metabolites that were similarly affected by space flight and ageing are not enough. The authors should explore more metabolites. A comprehensive comparison of metabolites that are affected by space flight and ageing would be important.

We thank the reviewer for this important comment. We agree with the comment that showing comprehensive comparison of metabolites is important to support our conclusion and have showed results of all 40 metabolites examined by NMR-based metabolome analyses in Supplementary Figures 3 and 4.

We also have searched for plasma metabolites, which changed solely by the space flight

or changed by both the space flight and Nrf2-deficiency to the same direction. We found six metabolites (i.e., glycerol, glycine, succinate, glutamine, carnitine and formate), which suffice one of the two criteria. Of note, among the six metabolites, the plasma levels of glycerol, glycine, and succinate were changed to the same direction with the changes observed in human ageing. In contrast, while plasma levels of glutamine, carnitine and formate changed significantly by the space flight, the levels either changed moderately (glutamine and carnitine) or to the reverse-direction in the human ageing analysis. We show the data of the mouse analyses and the human cohort analyses for the former three metabolites in Figure 4, and the data of mouse analyses for the latter three in Supplementary Figure 3. The lack of associations of the latter three metabolites are explained clearly and succinctly in the text (Page 10, lines 11-12, 19-20 and page 11, line 1-4), and are shown below for reviewer and editor viewing (a-c and d-f indicate plasma metabolite levels in mice and human, respectively).

3. Does space flight also induce ageing-like changes of gene expression? The author can compare their RNA-seq data with some published ageing transcriptome data in mice.

We thank the reviewer for the constructive comments. In order to address this comment, we have compared the RNA-seq data with the published ageing transcriptome data. We would like to remind the reviewer that the relationship between space travel and ageing has been suggested by many publications, but only a limited level of transcriptome

analyses have been conducted before this study. To address the reviewer's comment, we have expanded our transcriptome analysis and the data are all shown in Supplementary Fig. 5.

We are very pleased to inform the reviewer the results of these analyses. Gene set enrichment analyses (GSEA) using gene set of aged mice (Enrichr) revealed that space-induced changes of gene expression were indeed enriched in ageing changes of the aged mice in the liver, TpB, BAT and WAT (please see Supplementary Fig. 5). These results suggest that the space stress induces ageing-like changes in gene expressions, as well as in metabolites (shown above). We have added sentences to explain this observation (page 11, line 8-12).

4. Besides the Nrf2 targets and some metabolic genes, what genes are affected by space flight in RNA-seq analysis? How many of them are dependent on Nrf2? Functional classification of these genes (such as GO term analysis) is also important. This will show the significance of Nrf2 during space flight adaptation.

We thank the reviewer for these scientific comments. We agree with the reviewer that providing comprehensive transcriptome data will help future space mouse and biology studies. To address the reviewer's comments, we have conducted extensive GSEAs using 50 hallmarked gene sets in MSigDB for the space flight-induced genes. We have deposited the transcriptome data to NCBI's Gene Expression Omnibus and we are also preparing a ToMMo-JAXA mouse database (tentative) for various space mouse data.

The results of these GSEA analyses suggest that in fact a number of pathways are affected by the space flight (Supplementary Table 1). This Supplementary Table 1 lists pathways that show significance of $P \leq 0.05$. For instance, 6 pathways that are downregulated and 10 pathways are shown in the Table that are upregulated in wild type FL mouse WAT compared with wild type GC mouse WAT, respectively. We have shown similar data from the analyses of Liver, WAT, BAT, Spleen, Thymus, Cbr, TpB, Mdb and Kidney. We have added sentences to explain this observation (page 8, line 2-3).

However, while we also have tried pathway analyses for Nrf2-dependent genes identified in Supplementary Fig. 1, unfortunately we could not find significant pathways. We surmise that further analyses with much deeper levels of interrogation in each tissue may be required to reveal details of Nrf2 function during the space flight adaptation.

REVIEWERS' COMMENTS:

Reviewer #1 (Remarks to the Author):

On my previous review I had only raised minor issues, which the authors have addressed in this revised version. Therefore, I support publication of the revised manuscript.

Reviewer #2 (Remarks to the Author):

All concerns are addressed.